# Hyoid Bone Metastases: An Unusual Case

**DOI:** 10.3390/reports6040059

**Published:** 2023-12-08

**Authors:** Gian Piero Di Marco, Cinzia Tucci, Enzo Iacomino, Vincenzo Corridore, Maria Lauriello, Alessandra Fioretti, Alberto Eibenstein

**Affiliations:** 1Department of Otolaryngology, San Salvatore Hospital, 67100 L’Aquila, Italy; gpdimarco@virgilio.it (G.P.D.M.); cinzia.tucci1@gmail.com (C.T.); enzoiacomino74@hotmail.com (E.I.); vincenzo.corridore@gmail.com (V.C.); 2Department of Biotechnological and Applied Clinical Sciences, University of L’Aquila, 67100 L’Aquila, Italy; lauriellomaria@gmail.com (M.L.); alberto.eibenstein@univaq.it (A.E.); 3ENT Unit, European Hospital, 00149 Rome, Italy

**Keywords:** hyoid bone, metastasis, adenocarcinoma, neck tumefaction, neck surgery

## Abstract

(1) Background: Secondary tumors of the hyoid bone are extremely rare in clinics. In the literature, there is only one study about hyoid bone metastases from sigmoid adenocarcinoma. (2) Methods: We report a case of hyoid bone metastases in a 78-year-old patient treated for rectum and sigmoid colon adenocarcinoma. (3) Results: A mass excision surgery of a rounded osteolytic mass of 4.5 × 3.6 cm in size in the central part of the hyoid bone was performed under general anesthesia, according to the multidisciplinary tumor board recommendation. (4) Conclusions: Hyoid bone metastases can occur in the rectum and sigmoid colon adenocarcinoma. A total body bone scintigraphy and CT examination are suggested to detect silent bone metastases in patients with a history of cancer and neck masses. The prognosis is good, but a regular follow-up is recommended.

## 1. Introduction

The hyoid bone works as a site of attachment for pharyngeal muscles, the tongue and neck muscles and has a key role in coordinating swallowing and speech. Hypopharyngeal cancers include tumors arising between the hyoid bone and the lower limit of the cricoid cartilage. Hypopharyngeal cancers can be located in the post-cricoid, the piriform sinus and the posterior pharyngeal wall and are characterized by local and lymphatic spread. In total, 70% of patients with hypopharyngeal cancers present lymph node involvement at the time of diagnosis. Squamous cell carcinoma is the most common histology, while adenocarcinoma, sarcoma and non-epidermoid carcinoma occur in 5% of cases. The size and location of primary hypopharyngeal cancer determine the symptomatic severity. The most common presenting symptoms of hypopharyngeal cancer are bleeding, pain and dysphagia. Advanced hypopharyngeal cancer can invade the larynx, which consequently compromises the airway and aspiration. Primary tumors of the hyoid bone are rare, and they can be presented in the form of a benign osteoma, osteosarcoma, plasmacytoma, chondrosarcoma, giant cell tumor and aneurysmal bone cyst. Surgical management depends on the site of the tumor, and it requires a combination of partial or total pharyngectomy and laryngectomy. Hyoid resection can result in dysphagia, given the destabilization of the neck muscles, but in selected patients, can be well tolerated.

Secondary tumors of the hyoid bone are extremely rare in clinics. The larynx, vallecula and pyriform sinus are the primary malignancies that frequently metastasize to the hyoid bone because of their proximity [1].

In the literature, there are few cases of hyoid bone metastasis from distal primary malignant tumors, such as renal cell carcinoma, breast cancer and liver carcinoma [2,3,4]. Hyoid bone metastasis from lung adenocarcinoma [5], melanoma [6] and sigmoid adenocarcinoma [7] are also reported in the literature but are exceedingly rare.

Patients with hyoid bone tumors usually present with progressive dysphagia first for solids and then for liquids as the tumor becomes more advanced, and they may have a palpable neck mass [8]. An examination of the neck and a fiber-optic examination of the hypopharynx and larynx are crucial in the examination of patients with suspected hypopharyngeal cancers or hyoid bone metastasis. Hyoid bone metastasis can be demonstrated with contrast-enhanced computed tomography (CT) of the neck, chest, and abdomen, looking for a loco-regional spread. Distant involvement not identified on CT scans can be detected by total-body bone scintigraphy and a positron emission tomography (PET) scan. Total body bone scintigraphy and PET combined with CT help to locate occult disease elsewhere in the body. A multidisciplinary team approach with a head and neck surgeon and medical oncologist is recommended for the treatment of hyoid bone tumors and metastasis.

In this case report, we present a patient with hyoid bone metastases previously treated for rectum and sigmoid colon adenocarcinoma.

## 2. Detailed Case Presentation

We present the case of a 78-year-old Caucasian male patient with a 5-month history of non-specific worsening dysphagia. The patient was a carrier of ileostomy after extensive demolition surgery for a rectum and sigmoid colon adenocarcinoma, which was diagnosed and treated seven years before. After surgery, the patient underwent chemoradiotherapy. From the patient’s history, we noted that the patient was affected by myelopathy, arterial hypertension, hiatal hernia and benign prostatic hypertrophy.

A physical examination revealed a discrete tumefaction in the middle of the anterior neck region, without pain or tenderness. The mass had a stretched elastic consistency and quite a tense regular surface; it had an unclear boundary, and it was fixed to surrounding structures. The skin overlying the tumefaction was normal, and the neck was clinically negative.

A rounded osteolytic mass of 4.5 × 3.6 cm in size in the central part of the hyoid bone was evident from a contrast-enhanced computed tomography (CT).

CT demonstrated massive osteolysis and cortical destruction with the enlargement of the body of the hyoid bone. The osteolytic mass of the hyoid bone showed a slight soft tissue extension toward the pre-epiglottic region. An evident extension towards pre-hyoid muscles was also noted. The lumen of the airways was not affected, and no pathological changes were evident in the surrounding structures of the neck (Figure 1, Figure 2 and Figure 3).

In a total-body bone scintigraphy, the accumulation of 99mTc was not detected in the anterior neck but only in the left X rib, which was the site of a previous fracture (Figure 4).

Given the uncertain nature of the mass, we performed ultrasound-guided fine-needle aspiration cytology (FNAC), and the result was reported as a metastatic tumor.

After pathological diagnosis, mass excision surgery was performed under general anesthesia, according to the multidisciplinary tumor board recommendation. Informed consent was obtained from the patient. Surgery consisted of the hyoid resection of the tumor enlarged to suprahyoid and subhyoid muscles. Afterward, a tracheotomy and installation of the Portex-type tracheal cannula were performed, and a nasal feeding tube was inserted (Figure 5).

The macroscopic examination of the excised neoformation revealed a tumor mass with diameters of 4.7 × 5.2 × 3.0 cm, including the middle segment of hyoid bone 4.0 cm in size. The histopathologic examination identified a moderately differentiated adenocarcinomatous tissue with undamaged margins. The immunohistochemical examination showed an expression of Cytokeratin 20 and CDX2, while immunostainings for Cytokeratin 7 and TTF1 were negative (Figure 6). The cortex of the hyoid bone was infiltrated. The resection margins were undamaged. Based on the above findings, a diagnosis of metastatic colon adenocarcinoma was provided.

During the postoperative period, the patient received chemotherapy treatment. In that period, he also had a bone scintigraphy and an 18-F-FDG PET/CT. (Figure 7).

The patient’s postoperative evolution was favorable: no recurrence was observed after the corresponding chemotherapy. The patient returned for regular ENT evaluation without signs of loco-regional recurrence or metastasis at 6 and 12 months follow-up postoperatively.

## 3. Discussion

A not painful tumefaction of the middle anterior neck region is common in clinics and, above all, in children or young adults. It usually is a benign mass of hyoid bone that originates from the thyroglossal duct, and it can still be found in adults with atypical symptoms [9], lymphadenopathy by other causes such as acute infection, chronic inflammatory conditions or a benign lesion like a lipoma. Other causes of both cervical lymphadenopathy and dysphagia, like lymphoma, high esophageal cancer or other head and neck cancers, can be considered in the differential diagnosis. In the literature, there are a few cases of hyoid bone metastasis from distal malignant tumors, such as breast cancer, renal cell carcinoma, lung adenocarcinoma, melanoma and hepatocellular carcinoma [2,3,4,5,6]. Lung adenocarcinoma is more prone to vascular invasion and distant metastasis than squamous carcinoma [5]. Lung adenocarcinoma can also metastasize to the hyoid through the circulatory system, in particular, through the branch of the superior laryngeal artery. Any patient with a hyoid tumor and a previous history of melanoma should be considered in the differential diagnosis for metastatic melanoma [6]. Melanoma frequently metastasizes to cervical lymph nodes in the head and neck; bone metastases are observed in 11–17% of patients and usually occur in the axial skeleton [6]. The hyoid bone is also reported as a rare site of metastases in patients with renal cell carcinoma (RCC) [10] and in patients with rare lymphoproliferative disorders like solitary plasmacytoma (SP) and unicentric Castleman disease (UCD), characterized by a single set of locally enlarged lymph nodes [11]. A rare site of hyoid bone metastases in a patient with RCC is described and documented by ^1^8F-fludeoxyglucose (FDG)-positron emission tomography/a computed tomography (PET/CT) scan with metastases to bilateral lungs and multiple skeletal sites [10]. The osteolytic mass in hyoid bone suggestive of hyoid bone metastasis was confirmed using transaxial PET/CT and CT images with increased tracer uptake [10]. Zhang et al. described the case report of a man with a neck mass and a progressive foreign body sensation in the throat. The patient was evaluated with 18F-FDG positron emission tomography, which revealed focally increased radioactivity in the hyoid bone, and CT revealed osteolytic lesions [11].

Regarding our case report, in the literature, there is only another study about hyoid bone metastases from sigmoid adenocarcinoma [7].

Gastrointestinal adenocarcinoma is one of the most common malignant tumors in the world. Gastrointestinal adenocarcinoma usually spreads to the lungs and liver through the blood or lymph circulation, and regional lymph node metastases are also very common. In the literature, there have been reported cases of gastrointestinal adenocarcinoma metastasizing to the larynx and adjacent structures [12]. The most common metastatic laryngeal involvement is transglottic in 40% of cases, followed by supraglottic and subglottic in around 30% each and in less than 10% of the cases in the true vocal cords. The initial symptom in more than 60% is dysphonia, and the median time for laryngeal metastasis diagnosis is 3 years [12]. A case report of a patient with moderately differentiated sigmoid adenocarcinoma stage IV, distant metastasis at the time of presentation and disphonia due to the paralyzed right vocal cord with a left vocal cord compensation is described by Aljariri et al. [13].

Gastrointestinal adenocarcinoma metastases to unusual sites, such as paranasal sinuses, cryptorchid testis and in maxilla, with complaints of “swelling on the palate”, are also reported in the literature [14,15]. Oral adenocarcinoma metastases had a median survival of 6 months after diagnosis. A differential diagnosis between oral adenocarcinoma metastasis and squamous cell carcinoma, salivary gland tumor or pyogenic granuloma must be considered [15].

Regarding the primary location in colon cancer, we noticed the following variations in the metastasis site: sigmoid cancer had the highest rate of bone metastasis in the long-term follow-up compared to left and right colon cancer [16].

Colorectal adenocarcinoma could metastasize to the hyoid bone through the blood circulatory system, in particular through the lingual artery, the branch of the superior laryngeal artery or the branches of the lingual artery [17]^.^

In this case report, the hyoid bone mass of the patient was detected as an anterior tumefaction of the neck. A contrast-enhanced computed tomography (CT) showed a rounded mass of the hyoid bone with massive osteolysis and cortical destruction. After imaging, we included in the differential diagnosis only the diseases that may cause osteolytic lesions in the hyoid bone: multiple myeloma, primary bone tumors, could secondary tumors from different primary malignancies. Cytokeratin (CK) 20 and cytokeratin (CK) 7 are important markers for the diagnosis of metastatic tumors of gastrointestinal origin. As reported in the majority of well-differentiated or moderately differentiated intestinal adenocarcinomas, the CK7-negative and CK20-positive phenotypes confirmed the diagnosis of a metastatic tumor of gastrointestinal origin in our patient.

Negative total-body bone scintigraphy and the absence of disseminated disease misled us to misdiagnosis because we knew, from the literature, that unusual skeletal metastases of gastrointestinal adenocarcinoma occur more frequently in the case of liver and/or lung involvement than in patients without a disseminated disease [14,18]. In summary, osteolysis of the hyoid bone should be suspected of metastases in patients with a history of rectum and sigmoid colon adenocarcinoma.

The main role of imaging is to distinguish benign from malignant tumors and to assist in surgical planning. According to our case, because of the preoperative uncertain nature of the mass, a resection of the hyoid mass is recommended after the complete consideration of the patient’s physical status. A regular follow-up of the patient diagnosed with colon adenocarcinoma is mandatory.

## 4. Conclusions

Secondary tumors of the hyoid bone are extremely rare in clinics, so there is a global lack of experience in dealing with it, and, consequently, diagnosis and clinical management can be a challenge. Hyoid bone metastases can occur in the rectum and sigmoid colon adenocarcinoma, so in patients with a history of these carcinomas, the destruction of the hyoid bone could be suspected of metastases. Total body bone scintigraphy and CT examination are suggested to detect silent bone metastases in patients with a history of cancer and neck mass. A resection of the hyoid mass is strongly recommended. The prognosis of hyoid bone metastasis is good, but regular follow-ups are recommended.

## Figures and Tables

**Figure 1 reports-06-00059-f001:**
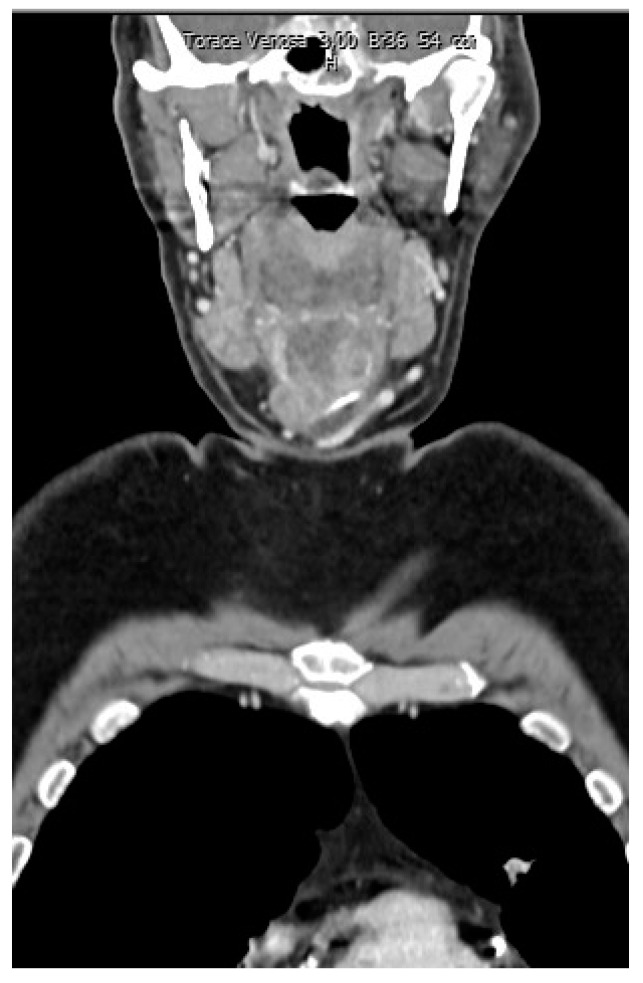
Contrast-enhanced computed tomography (CT) showed a rounded osteolytic mass of 4.5 × 3.6 cm in size in the central part of the hyoid bone: coronal view.

**Figure 2 reports-06-00059-f002:**
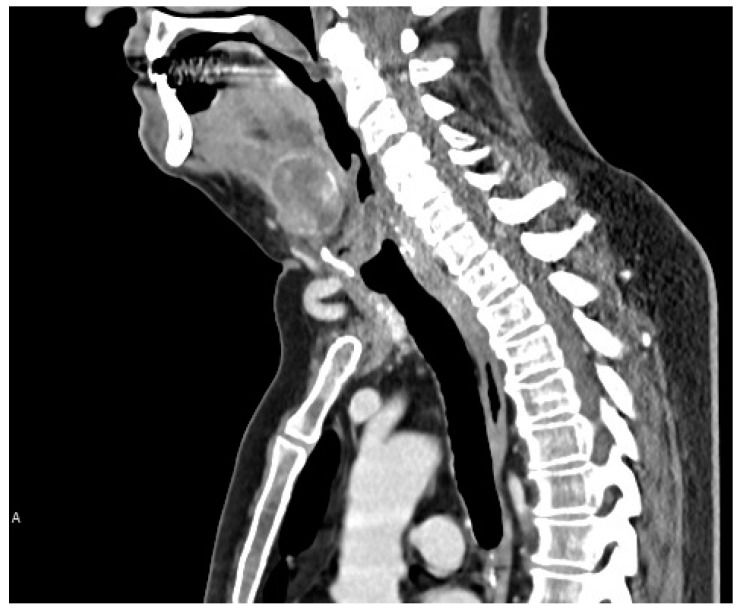
Contrast-enhanced computed tomography (CT) showed a rounded osteolytic mass of 4.5 × 3.6 cm in size in the central part of the hyoid bone: sagittal view.

**Figure 3 reports-06-00059-f003:**
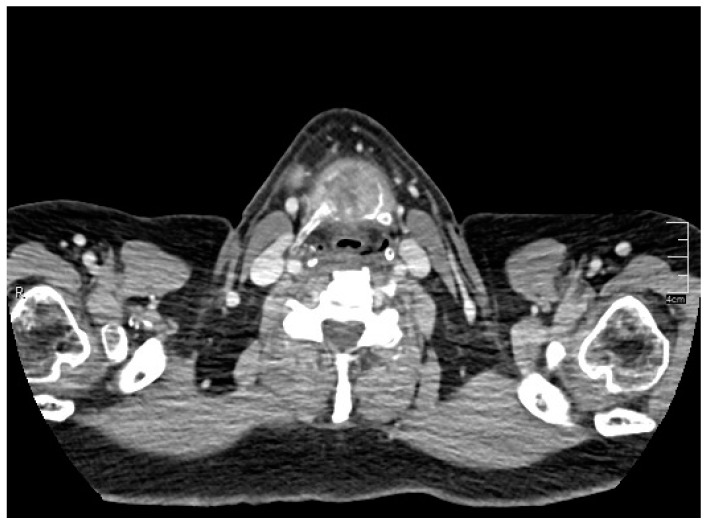
Contrast-enhanced computed tomography (CT) showed a rounded osteolytic mass of 4.5 × 3.6 cm in size in the central part of the hyoid bone: axial view.

**Figure 4 reports-06-00059-f004:**
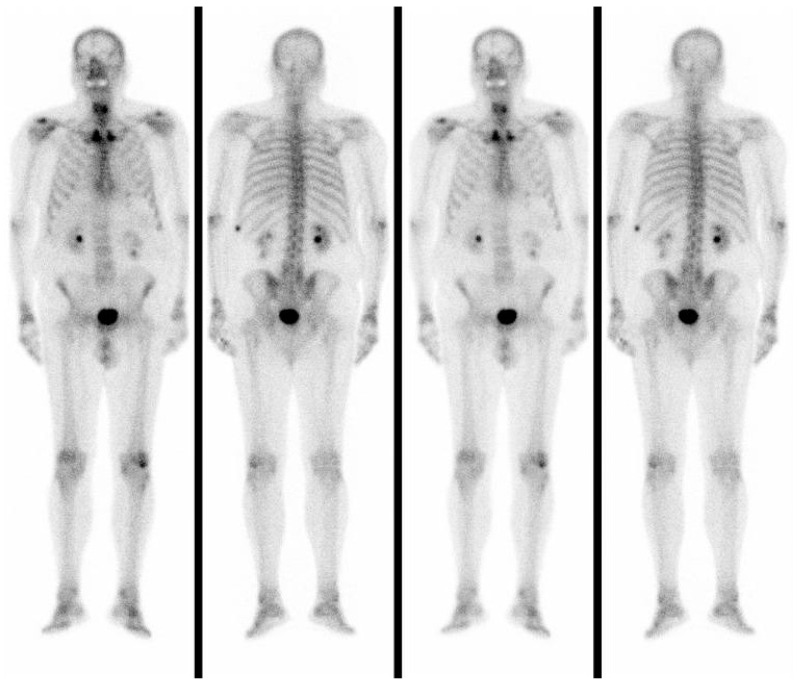
Total-body bone scintigraphy showing the accumulation of 99 mTc that was not detected in the anterior neck but only in the left X rib which was the site of a previous fracture.

**Figure 5 reports-06-00059-f005:**
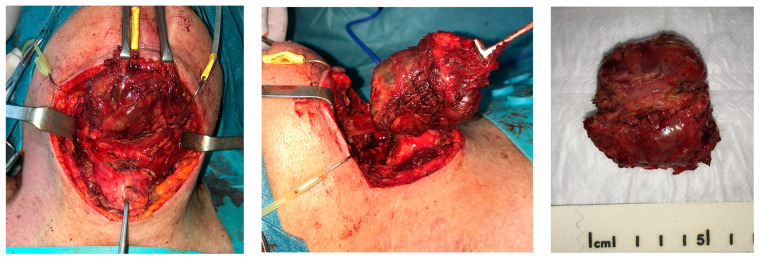
Hyoid resection of the tumor enlarged to suprahyoid and subhyoid muscles.

**Figure 6 reports-06-00059-f006:**
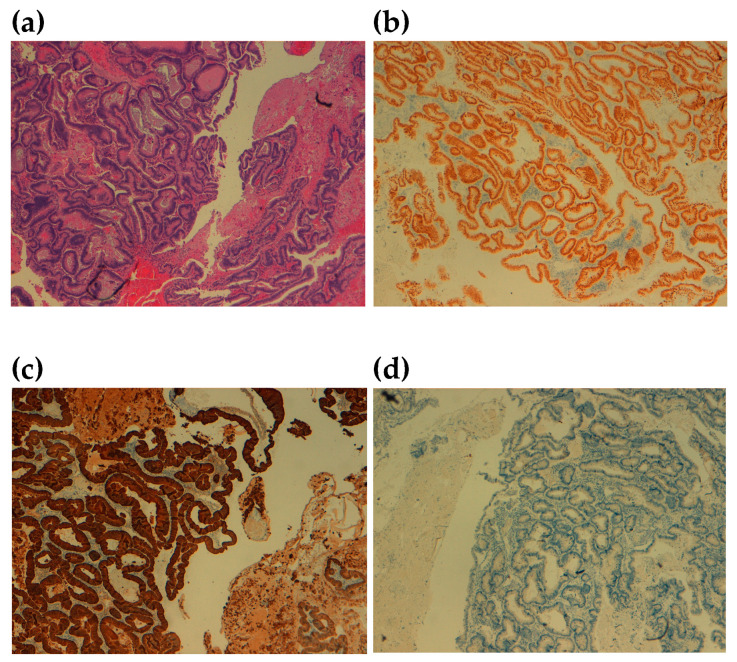
Histopathologic and immunohistochemical examination. (**a**) hematoxylin and eosin; (**b**) CDX2; (**c**) Cytokeratin 20; (**d**) Cytokeratin 7.

**Figure 7 reports-06-00059-f007:**
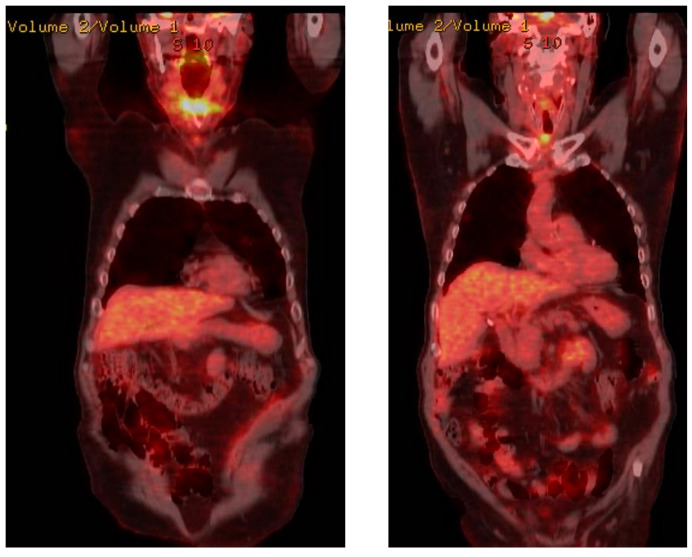
Postoperative 18-F-FDG PET/CT.

## Data Availability

No new data were created.

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
