# Peer review of "Hyoid Bone Metastases: An Unusual Case"

_reports, 2023, doi:10.3390/reports6040059_

Round 1

Reviewer 1 Report

Comments and Suggestions for Authors

Malignant tumors of the hyoid bone are rare and include sarcomas, plasmacytomas, and distant metastases. Only a few cases of tumors metastatic to the hyoid bone have been reported in the literature, including carcinomas originating in breast, kidney, liver, colon, and lung.

1. In this case report, authors present a patient with hyoid bone metastases previously treated for a rectum and sigmoid colon adenocarcinoma.

2. In literature, there are few cases of hyoid bone metastasis; sigmoid adenocarcinoma are also reported in the literature but exceedingly rare.

3. Radiological examination is suggested to detect silent bone metastases in patients with history of cancer and neck mass.

4. Secondary tumors of the hyoid bone are extremely rare in clinics, so there is a global lack of experience in dealing with it and, consequently, diagnosis and clinical management can be a challenge.

5. The conclusions are consistent with the evidence and arguments presented and they address the main question posed.

6. The references are appropriate.

7. Tables and figures are appropriate.

Author Response

Many thanks for your appreciation and time spent reviewing the article.

Reviewer 2 Report

Comments and Suggestions for Authors

For a case presentation the information presented is usefull. 

I suggest to introduce a picture of the histologic examination.(the staining performed to clarify the histopatological diagnosis)

Author Response

Many thanks for the time spent to review the article and for your advice. We added fig. 8 with the images of histologic examination and the staining performed as requested.

Reviewer 3 Report

Comments and Suggestions for Authors

The authors report a case of the hyoid bone metastases in a 78-year-old patient treated for rectum and sigmoid colon adenocarcinoma. The clinical case is well-described and well-presented. The paper is structured correctly. The clinical history is reported effectively. The images are accurate and appropriately represent the scope of the work. The discussion is comprehensive and detailed. The references are recent and up-to-date.

The conclusions clarify the therapeutic diagnostic process in the case of metastases to the hyoid bone, which can occur in rectum and sigmoid colon adenocarcinoma.

Author Response

(The authors gave the same response as above.)
